# Universal scaling relations for the rational design of molecular water oxidation catalysts with near-zero overpotential

Michael John Craig [1], Gabriel Coulter[1], Eoin Dolan[1], Joaquín Soriano-López [1], Eric Mates-Torres[1], Wolfgang Schmitt[1] & Max García-Melchor [1]*

A major roadblock in realizing large-scale production of hydrogen via electrochemical water splitting is the cost and inefficiency of current catalysts for the oxygen evolution reaction (OER). Computational research has driven important developments in understanding and designing heterogeneous OER catalysts using linear scaling relationships derived from computed binding energies. Herein, we interrogate 17 of the most active molecular OER catalysts, based on different transition metals (Ru, Mn, Fe, Co, Ni, and Cu), and show they obey similar scaling relations to those established for heterogeneous systems. However, we find that the conventional OER descriptor underestimates the activity for very active OER complexes as the standard approach neglects a crucial one-electron oxidation that many molecular catalysts undergo prior to O–O bond formation. Importantly, this additional step allows certain molecular catalysts to circumvent the "overpotential wall", leading to enhanced performance. With this knowledge, we establish fundamental principles for the design of ideal molecular OER catalysts.

[1] School of Chemistry, CRANN and AMBER Research Centres, Trinity College Dublin, College Green, Dublin 2, Ireland. *email: garciamm@tcd.ie

Reducing anthropogenic $CO_2$ emissions is an urgent challenge facing civilization in the 21st century[1]. Solar and wind energy are attractive carbon-free energy sources that could supply energy demand sustainably, but an important difficulty with reliance on these energy sources is their inherent intermittency and localized energy input. These two issues could be addressed by storing the excess energy in the form of chemical bonds, such as $H_2$, formed via electrochemical water splitting in an electrolyzer[2]. The hydrogen gas liberated at the cathode could then be stored, and eventually combined with oxygen in a fuel cell, providing an entirely renewable and clean energy supply with water as the only reaction product. Hydrogen produced via this method could also be utilized for the reduction of anthropogenic $CO_2$ to produce chemical feedstocks and hydrocarbon fuels which are easier to transport than $H_2$[3,4].

The bottleneck reaction in water splitting is the so-called oxygen evolution reaction (OER), occurring at the anode of the electrolyzer (Eq. (1))[5,6]. Catalysts which are cost-effective, highly active and robust for continued periods of time are yet to be found for this endergonic reaction, which demands large overpotentials ($\geq$400 mV) to reach substantial current densities ($\geq$10 mA/cm$^2$) and oxygen conversion rates[7,8].

$$2H_2O_{(l)} \rightleftharpoons O_{2(g)} + 4H^+ + 4e^- \quad E^0 = 1.23\,V \text{ vs. RHE} \quad (1)$$

The OER process typically occurs via four elementary steps involving different reaction intermediates and the formation of an O–O bond, which is eventually released as molecular oxygen. The two primary pathways proposed for the O–O bond formation are the water nucleophilic attack (WNA) and the interaction of two metal-oxo entities (I2M), as depicted in Fig. 1[9]. Both reaction paths begin with two proton electron transfer (PET) events[10], giving rise to a metal-oxo intermediate. In the case of WNA, the metal-oxo species subsequently undergoes the nucleophilic attack of a solvent water molecule and a further PET to generate the O–O bond, whereas in the I2M mechanism the O–O bond formation involves the coupling of two separate metal-oxo moieties.

With this understanding of the OER, Nørskov et al. showed that the activity of metal and metal oxide surfaces via a WNA mechanism can be readily assessed by computing the binding energies of the different reaction intermediates[11,12]. Particularly, the Gibbs energy change associated with each elementary step can be calculated as in Eqs. (2)–(5), where * denotes a metal active site, $a_{H^+}$ is the activity of the proton, $e$ is the number of electrons

involved, and $U$ is the applied potential:

$$H_2O + * \rightarrow HO^* + H^+ + e^-$$
$$\Delta G_1 = \Delta G_{HO^*} - eU + k_B T \ln a_{H^+} \quad (2)$$

$$HO^* \rightarrow O^* + H^+ + e^-$$
$$\Delta G_2 = \Delta G_{O^*} - \Delta G_{HO^*} - eU + k_B T \ln a_{H^+} \quad (3)$$

$$O^* + H_2O \rightarrow HOO^* + H^+ + e^-$$
$$\Delta G_3 = \Delta G_{HOO^*} - \Delta G_{O^*} - eU + k_B T \ln a_{H^+} \quad (4)$$

$$HOO^* \rightarrow O_2 + H^+ + e^-$$
$$\Delta G_4 = \Delta G_{O_2} - \Delta G_{HOO^*} - eU + k_B T \ln a_{H^+} \quad (5)$$
$$= 4.92\,eV - \Delta G_{HOO^*} - eU + k_B T \ln a_{H^+}$$

By applying the computational hydrogen electrode model to the above expressions[13], the chemical potential of the $H^+/e^-$ pair can be replaced by one-half of the chemical potential of molecular hydrogen, as these two species, by definition, are in equilibrium at 0 V versus the reversible hydrogen electrode (RHE). Furthermore, the introduction of the error arising from the modeling of the $O_2$ molecule with density functional theory (DFT) methods can be evaded by fixing the overall Gibbs reaction energy, $\Delta G_{O_2}$, to the experimental value of 4.92 eV.

From the relative Gibbs energies in Eqs. (2)–(5), one can then determine the potential limiting step (PLS), defined as the most thermodynamically demanding step:

$$\Delta G_{PLS} = \max\{\Delta G_i\} \quad (6)$$

And the theoretical overpotential by subtracting the thermodynamic redox potential of water:

$$\eta_{theor} = \left(\frac{\Delta G_{PLS}}{e}\right) - 1.23\,V \quad (7)$$

It is important to note that $\eta_{theor}$ does not lead to a direct prediction of the current density attained at that given potential, but rather an upper bound of the voltage at which OER activity is to be initiated. Hence, an ideal catalyst with $\eta_{theor} = 0$ V would perfectly distribute the overall change in Gibbs energy throughout the four OER steps, i.e. $\Delta G_{1-4} = 1.23$ eV.

Further insight into the energetics of the OER intermediates was provided by Koper[14], showing that the HOO* and HO* binding energies ($\Delta G_{HOO^*}$ and $\Delta G_{HO^*}$) on metal surfaces scale linearly with a constant energy difference of ca. 3.2 eV. From this OER scaling relation, the following important implications emerged. Firstly, the value of $\Delta G_{HOO^*}$ can be directly obtained from the calculated $\Delta G_{HO^*}$, and vice versa, which reduces the computational time needed to assess the OER activity of a given surface catalyst. Secondly, because the energy difference between the HOO* and HO* intermediates is larger than the expected value for an ideal catalyst ($2 \times 1.23 = 2.46$ eV), the best possible material subject to this constraint would have a $\Delta G_{O^*}$ value exactly halfway between the $\Delta G_{HOO^*}$ and $\Delta G_{HO^*}$ binding energies, resulting in a minimum theoretical overpotential of 370 mV ([3.2–2.46 eV]/2). Consequently, OER electrocatalysts are predicted to be limited by an "overpotential wall" of 370 mV, in excellent agreement with the experimental benchmark studies of state-of-the-art electrocatalysts[8]. While the uncertainty associated with the calculation of $\Delta G_{HOO^*} - \Delta G_{HO^*}$ can reduce that overpotential wall (see Supplementary Fig. 6 and Supplementary Note 3 for a detailed discussion)[15], as has been experimentally observed for a few exceptional OER electrocatalysts[16–21], hurdling or reducing this wall remains a major challenge.

Another major implication of the above OER scaling is that since $\Delta G_1$ and $\Delta G_4$ are rarely PLS, the Gibbs energy difference between the O* and HO* intermediates ($\Delta G_{O^*} - \Delta G_{HO^*}$) can be

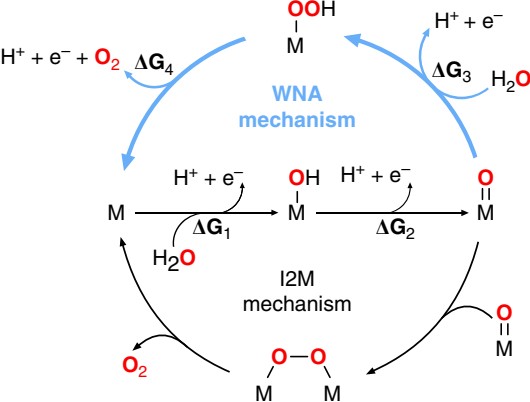

**Fig. 1** Catalytic cycles for the two primary reaction pathways proposed for the OER

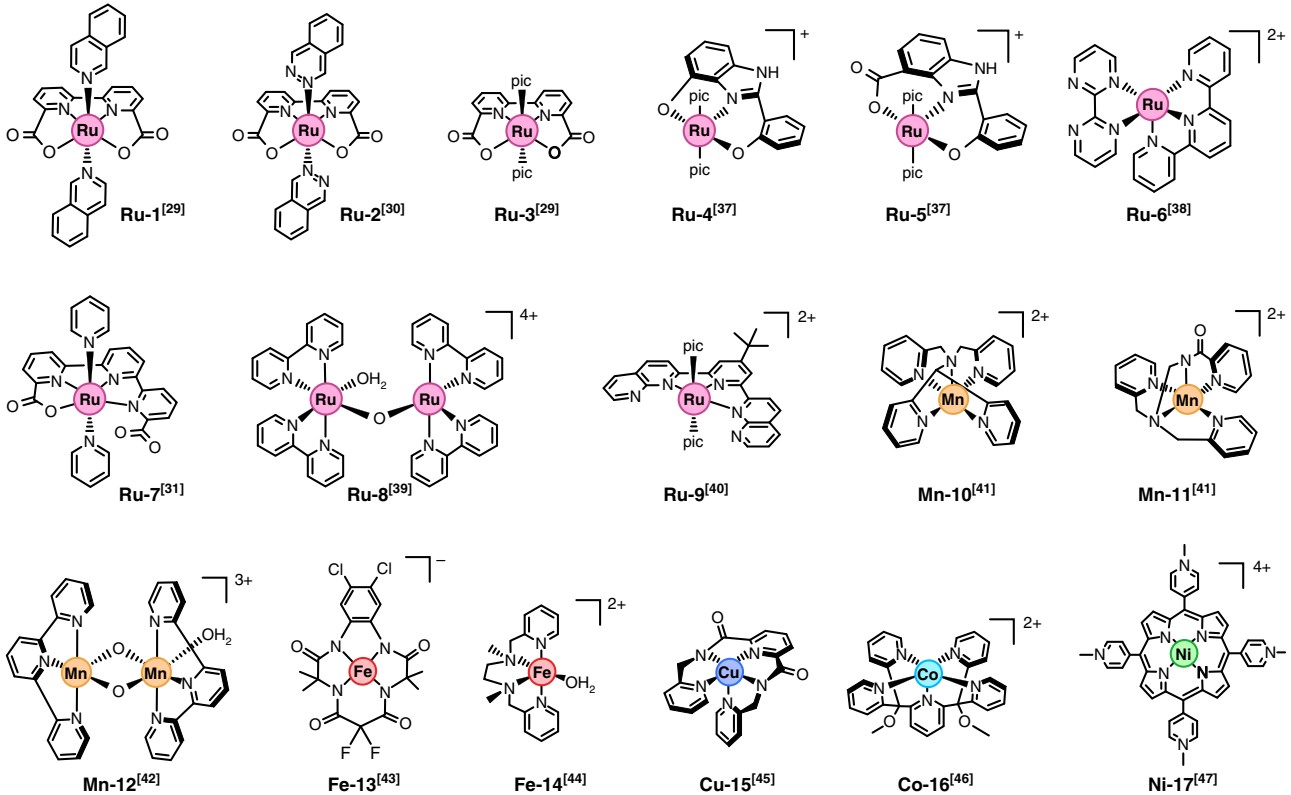

**Fig. 2** Molecular OER catalysts investigated in this work, taken from refs. [29-31,37-47]. Chemical structures correspond to the vacancy intermediate in the OER mechanisms depicted in Fig. 1. The 4-picoline ligands in **Ru-3-5** and **Ru-9** have been replaced by "pic" for clarity

employed as a reaction descriptor to predict the OER activity of heterogeneous catalysts[9,22]. In fact, this descriptor has been successfully applied to rationalize the activity of a wide variety of metal oxide and single-atom electrocatalysts, wherein the most active ones exhibit a descriptor which is close to the optimal value predicted by the scaling relations, i.e. $3.2/2 = 1.6$ eV[15,23,24]. This descriptor also allows for the fast, high-throughput screening of materials from ab initio simulations, a procedure that has proven to be effective in the design of more efficient electrocatalysts for the other relevant reactions such as the hydrogen evolution reaction and $CO_2$ reduction[25,26].

Recent theoretical studies have reported OER scaling relations for a few model molecular systems bearing a corrole, porphyrin ligands, and functionalized graphitic materials[15,27,28], but the generalization of these relations to catalysts featuring different metals and ligand scaffolds, as well as the applicability of the OER descriptor, is yet to be demonstrated. Herein we confirm the existence of universal scaling relations for a wide variety of well-established molecular OER systems. This investigation is prompted by the superior performance of certain homogeneous catalysts for which turnover frequency values (TOFs) of 2–3 orders of magnitude higher than the best heterogeneous systems have been reported[29-31]. To demonstrate the robustness of the established scaling relations, we have exhaustively interrogated 17 selected molecular OER catalysts featuring a wide variety of metals and ligands (Fig. 2)—all of them experimentally tested and reported in the literature—some of which are amongst the most active OER systems known to date. For catalysts with two potential active sites, only one of them was considered for simplicity. In selecting catalysts, we considered only those exhibiting a well-defined and well-established structure. For instance, Ir catalysts bearing pentamethylcyclopentadienyl (Cp*) ligands[32,33]

were discarded, despite some of them showing considerable OER activity, as the integrity of many of these complexes has been shown to be compromised under OER conditions[33,34]. More complex and unique systems with multimetallic centers were also discarded due to the multiple distinct valence states and active sites which would have to be considered prior to oxygen evolution[35].

Interestingly, we find that the best molecular OER catalysts can circumvent the scaling between the HOO* and HO* intermediates by undergoing an additional one-electron oxidation prior to O–O bond formation. As a result, we predict molecular catalysts enabling this extra oxidation to potentially behave as ideal OER catalysts, exhibiting zero theoretical overpotential. Further, we demonstrate that the activity of such highly active systems is underestimated by the conventional OER descriptor and propose a new descriptor to screen and narrow down the search of promising molecular catalysts. Finally, we use all this knowledge to establish fundamental principles for the rational design of ideal OER catalysts to advance the development of commercial water electrolyzers.

## Results and discussion

**OER scaling relations for state-of-the-art molecular OER catalysts.** Firstly, we set out to prove the scaling relation between the HO* and HOO* intermediates observed for heterogeneous systems using the molecular OER catalysts depicted in Fig. 2. With this aim, we employed DFT calculations at the TPSSh level along with Grimme D3 dispersion corrections to obtain the Gibbs energies of the different OER intermediates, following the methodology described in the "Computational Methods" section and applying the computational hydrogen electrode model,

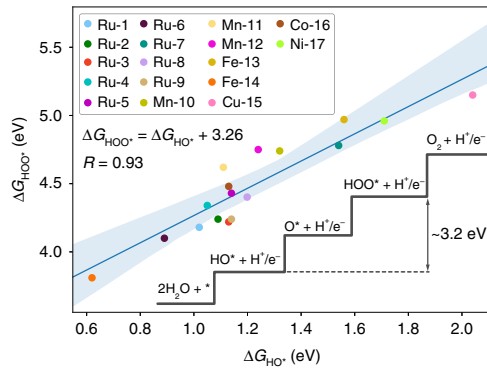

**Fig. 3** Linear scaling relation between the HO* and HOO* intermediates for the molecular OER catalysts investigated in this work and depicted in Fig. 2. The shaded blue region represents a 99% confidence interval for the linear model

previously used in studies of heterogeneous oxide catalysts for the OER[22].

As shown in Fig. 3, the representation of the binding energies of the HOO* species ($\Delta G_{HOO^*}$) against those obtained for the HO* species ($\Delta G_{HO^*}$) shows that the energetics of these two intermediates are strongly related, displaying a linear correlation with a slope close to unity—as expected based on the bond order conservation principle[36]—and an intercept of 3.26 eV. This confirms that the same linear scaling relation observed in metal and metal oxide systems applies to any molecular OER catalyst. Also noteworthy is the robustness of this relation, which is derived from complexes showcasing many distinct ligands and metal centers[29–31,37–47]. In addition, the constraint imposed by the OER scaling implies that both homogeneous and heterogeneous catalysts are subject to a minimum overpotential of ca. 370 mV. The OER scaling relation can therefore now be labeled as "universal", since it can be applied to any OER material regardless of their nature.

We then examined in detail the data presented in Fig. 3 to identify catalyst features which could improve the OER scaling, thus reducing the predicted overpotential. Particularly, we hypothesized that intramolecular H-bonds between the HOO* group and the metal ligands might offer that opportunity given the innate rigidity of the HO* group (Supplementary Fig. 1). This additional stabilization of the HOO* intermediate would shift down $\Delta G_{HOO^*}$ with respect to $\Delta G_{HO^*}$, closing the energy gap between the two intermediates, and therefore, improving the OER scaling. To prove our hypothesis, we classified our data points as catalysts with or without a H-bond by measuring the minimum distance from the H atom in the HOO* group to the nearest N/O-ligand using a number of different cut-offs, chosen such that each split contained sufficiently different data points. As we anticipated, the ability of the HOO* intermediate to form intramolecular H-bonds decreases the value of $\Delta G_{HOO^*}$ and, consequently, reduces the energy difference $\Delta G_{HOO^*} - \Delta G_{HO^*}$. This effect is conveyed in Supplementary Fig. 1, where two clear distinct scaling relations are obtained depending on whether the HOO* intermediate exhibited a H-bond or not. Importantly, we note that all linear trends display slopes close to unity; however, those catalysts with an intramolecular H-bond present in the HOO* intermediate show a value of $\Delta G_{HOO^*} - \Delta G_{HO^*}$ that is ca. 0.1 eV lower than those without the H-bond. While this effect might not be decisive—being within our reported mean absolute error (MAE) of the chosen functional—we conclude that H-bonding can be leveraged to improve the scaling, and therefore, it should be considered as an important feature in the design and fine-tuning of molecular OER catalysts. H-bonding can also be

leveraged to boost the kinetics of the O* to HOO* step in the WNA mechanism, as recently shown by experiments[31], while theoretical investigations have highlighted the possibility that the second coordination sphere can modulate the binding energy of the crucial O* intermediate[48].

We note that molecular dynamics simulations including explicit solvation would allow for a more complete characterization of the significance of intramolecular H-bonds on the molecular OER scaling. This methodology has been used to explain the mediation of O–O bond formation by solvent structures for the Ru$^V$–oxo intermediate of Ru-7 elsewhere[49,50]. Applying this level of analysis to all of our investigated catalysts (and OER intermediates) is, however, extremely computationally demanding. To circumvent this, DFT calculations in concert with minima-hopping have been employed to study solvent effects on the OER intermediates adsorbed on rutile IrO$_2$, showing a larger stabilization of the HOO* intermediate on a O-covered surface[51]. Such investigations, however, are outside the scope of this work, which is attempting to establishing and generalizing OER-scaling relations for molecular catalysts and to set the guidelines for the accelerated discovery of high-performance OER catalysts using methods that are computationally efficient and less time consuming.

**OER volcano plots and reaction descriptors**. After demonstrating that molecular catalysts hold a scaling relationship between the HO* and HOO* intermediates, we tested whether the OER descriptor proposed for heterogeneous catalysts, i.e. $\Delta G_{O^*} - \Delta G_{HO^*}$, could also be applied to rationalize the experimental performance of the investigated complexes. By plotting this descriptor against the computed theoretical overpotential using in Eqs. (2)–(7), a volcano representation is obtained, where the most active catalysts should lie at the top. This representation also allows us to categorize molecular catalysts based on the PLS. More specifically, catalysts lying on the left-hand side of the volcano display a small OER descriptor value, and therefore, the O* to HOO* transition becomes the PLS. On the other hand, catalysts found on the right leg of the volcano exhibit a high descriptor value, which dictates the OER overpotential.

The volcano plot obtained for the 17 molecular catalysts is shown in Fig. 4a, where, surprisingly, none of them are predicted to have an outstanding OER activity. This contrasts starkly with the reported experimental data, which have proven some of the complexes to be among the best molecular OER catalysts. In the following, however, we demonstrate that this unsuspected behavior is caused by an oversimplification of the reaction mechanism, typically assumed in most theoretical studies of heterogeneous OER catalysts. Notably, some metals, such as Ru, Mn, or Fe, have been experimentally proven to undergo an additional one-electron transfer (ET) before O–O bond formation[52–55]. Bearing this in mind, we computed (whenever possible) the Gibbs energies for the ET from the M(IV)-oxo to M(V)-oxo species with M = Ru, Mn, Fe, hereafter referred to as O(IV)* and O(V)*, respectively (see Supplementary Tables 2 and 3 for details). The Gibbs energy change associated with this step, and the subsequent O–O bond formation via a PET event, are given by

$$O(IV)^* \rightleftharpoons O(V)^* + e^-$$
$$\Delta G_{3'} = \Delta G_{O(V)^*} - \Delta G_{O(IV)^*} - eU \quad (8)$$

$$O(V)^* + H_2O \rightleftharpoons HOO^* + H^+$$
$$\Delta G_{4'} = \Delta G_{HOO^*} - \Delta G_{O(V)^*} + k_B T \ln a_{H^+} \quad (9)$$

Interestingly, upon accounting for the additional ET in the M(IV)-oxo intermediates, we observed that catalysts which undergo this step exhibit an improved theoretical overpotential, deviating

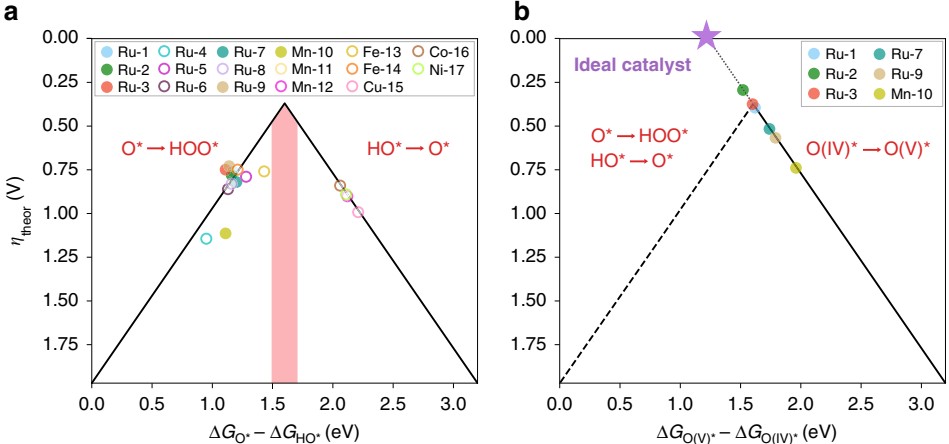

**Fig. 4** 2D-volcano representations featuring different OER descriptors. **a** Volcano plot using the conventional OER descriptor. Full markers denote catalysts which are theoretically predicted to undergo an additional ET step before O–O formation (see Supplementary Table 3). The red shaded area indicates the optimal range for the conventional OER descriptor. **b** Volcano plot including only the molecular catalysts predicted to undergo an additional ET. In this volcano, the calculated Gibbs energy for the additional ET from the M(IV)-oxo to M(V)-oxo intermediate is represented in the x-axis as a new OER descriptor. Note, the same scale and the left leg of the volcano in **a** has been used for comparability. The dotted line in **b** is drawn to guide the eye towards an ideal OER catalyst. The potential limiting step on each side of the two volcano plots is indicated in red text

significantly from the lines defined by the volcano in Fig. 4a. This divergence from the scaling stems from the fact that those catalysts are neither limited by the HO* to O*, nor the O* to HOO* steps. Instead, the additional ET from the M(IV)-oxo species becomes the PLS, which demands less energy leading to a lower predicted overpotential. Based on this observation, we proceeded to apply the Gibbs energy change associated with the additional ET step, $\Delta G_{O(V)*} - \Delta G_{O(IV)*}$, as an alternative OER descriptor on the x-axis. The result of adopting this new descriptor can be appreciated in the volcano plot depicted in Fig. 4b, where the relevant catalysts now lie on the right leg of the volcano and where the best Ru-based catalysts cluster together around the top. Notably, DFT calculations predict **Ru-1–3**, **Ru-7**, and **Ru-9** to exhibit the lowest theoretical overpotential amongst the 17 complexes investigated in this work, which agrees with the highest experimental TOFs reported for these complexes (see Supplementary Fig. 5 and Supplementary Note 2). It is important to note that the active species for **Ru-9** has come under scrutiny[56], but our calculations predict that this catalyst would be a relatively active OER catalyst. The trend in the experimental TOFs (see Supplementary Table 4) is also captured by the new descriptor, highlighting its predictive (both qualitatively and semi-quantitatively) power and its potential for the rapid screening of molecular OER catalysts. We also note that when comparing energy differences between the PET and one-electron oxidation steps, it is important to consider whether this difference is within the inherent DFT error (Supplementary Table 1), in which case neither of the two pathways can be entirely discarded.

Another important observation from Fig. 4b is that **Ru-1** and **Ru-2** are predicted to display a theoretical overpotential below 370 mV, suggesting that molecular catalysts undergoing the O(IV)* to O(V)* transition might be able to hurdle the overpotential wall. In fact, such a catalyst with the perfect distribution of the energy levels would present a Gibbs energy of only 1.07 eV for each of the steps in Eqs. (3), (8), and (9)—obtained by dividing the energy difference between the HO* and HOO* intermediates imposed by the conventional scaling, i.e. 3.2 eV, between the three steps. Assuming that one step between the HO* and HOO* intermediates equals the thermodynamic potential of water (i.e. 1.23 eV), the remaining two steps would sum to 1.97 eV (3.2–1.23). This implies that the design of ideal

OER catalysts with 0 mV overpotential is theoretically within reach, provided that the 1.97 eV is properly distributed across those two steps. With this enlightening insight, we next examined the optimal range of binding energies which would lead to a near-zero or 0 mV overpotential using the 3D volcano representations in Fig. 5. We note that while the oxidation of O(IV)* to O(V)* has been known to be of critical importance experimentally and theoretically for individual catalysts[57], this is the first computational rationalization of exceptional catalyst activity, using this intermediate, across a set of active catalysts.

The difference between the 3D volcanos derived for the catalysts following the conventional 4-PET mechanism (Fig. 5a) and that with the additional ET step (Fig. 5b) is striking, namely, the area of high activity is approximately two times larger for the latter volcano, while the activity itself is clearly much more favorable (i.e. $\eta_{\mathrm{theor}}$ values of 370 and 0 mV, respectively). The shape of the volcano featuring the new OER descriptor can be rationalized by considering the equations that determine the boundaries separating the distinct regions (see Supplementary Fig. 2 and Supplementary Note 1), which denote the PLS at each region, as shown in Fig. 5b. This new volcano provides also a comprehensive illustration of the observed superior performance of Ru-based OER catalysts compared with other metal-based catalysts. Furthermore, it sets up a general guideline for the future design of complexes as cost-effective and high-performance OER catalysts, given the propensity of some earth-abundant metals (e.g. Mn and Fe) to stabilize high-valent oxo intermediates[58–60].

**Comparing WNA and I2M reaction mechanisms under OER conditions.** Our investigations have thus far assumed a WNA mechanism. However, it is important to note that even if the I2M pathway—or indeed any other mechanism—was found to be more favorable, it would only result in a better-predicted activity. Hence, the scaling relations and reaction descriptors established in this work are perfectly valid and relevant, since they provide an upper limit of the OER activity for a given molecular catalyst, requiring only a few DFT calculations. Besides, when comparing the WNA and I2M mechanisms, one should bear in mind that the preference for one or the other pathway will be strongly dependent on the reaction conditions, something that can be overlooked. In particular, the O–O bond formation via the WNA

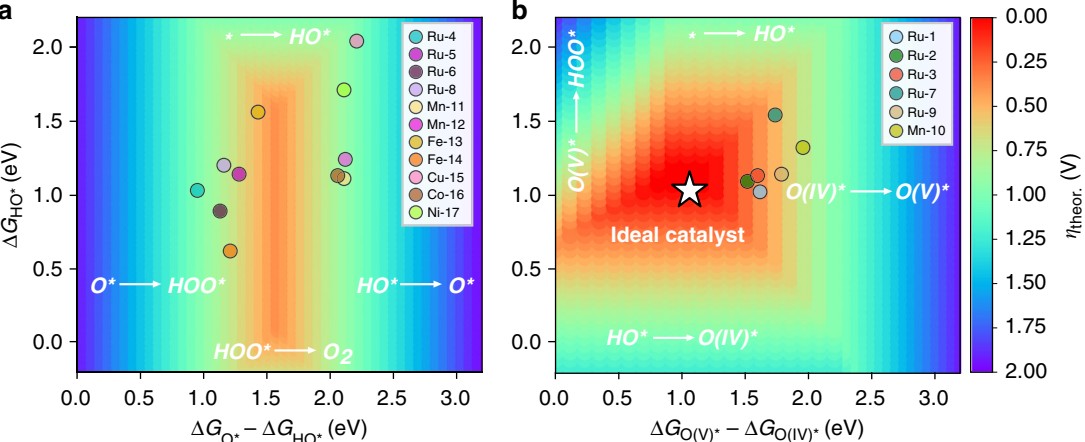

**Fig. 5** Three-dimensional volcano representations including. **a** OER catalysts following the conventional 4-PET pathway and **b** catalysts theoretically predicted to undergo the additional ET step before O–O formation. The PLS on each region of the volcano plots is indicated

mechanism involves a PET step, and therefore, the Gibbs energy associated with this process will be reduced as potential increases according to Eq. (4). On the contrary, the formation of the M–O–O–M dimer through the I2M mechanism is a chemical step, implying that the Gibbs energy with respect to the monomer will remain invariant with the applied bias. To illustrate this, we have analyzed this mechanism for **Ru-1, 2,** and **3** for which the corresponding Ru(V)-oxo species has been kinetically confirmed to dimerize leading to the complex Ru(IV)–OO–Ru(IV)[29,30]. Furthermore, anchoring **Ru-1** on indium tin oxide has also been shown to increase the observed overpotential, presumably due to a change in mechanism from I2M to the WNA[61]. It should be noted that the optimized geometries of the complexes as single molecules or dimers is octahedral, despite the fact that the favorable activity of some Ru catalysts have been explained by their ability to dynamically access a seven-coordinated intermediate[62]. The effect of the applied voltage on the thermodynamics of both WNA and I2M mechanisms for **Ru-1** is summarized in Fig. 6, while the data for **Ru-2** and **3** dimers is presented in Supplementary Table 3.

As can be observed from Fig. 6, DFT calculations predict that the I2M mechanism will dominate at applied voltages up to 1.98 V, in good agreement with the reported experiments using cerium ammonium nitrate as the sacrificial chemical oxidant ($E^0 = 1.61$ vs. NHE). At more oxidizing potentials, however, simulations predict the WNA pathway to become accessible. Hence, the I2M mechanism is expected to have an important contribution to the observed OER activity at low to moderate potentials (or mild chemical oxidants), whereas the WNA is expected to compete at strong oxidizing conditions. As a result, the I2M path should be considered for catalysts with low predicted overpotentials, especially if the O* to HOO* transition is the PLS. The results presented in Supplementary Table 3 indeed suggest that the degree of exergonicity of this dimerization step is the deciding factor in the improved kinetics of **Ru-1** with respect to **Ru-2** and **3**. The importance of circumventing the overpotential wall through O–O-based methods, such as the I2M mechanism has been discussed previously[15].

We emphasize that a complete quantum chemical characterization of the underlying mechanism for catalysts is to remain vital due to the need for increased accuracy in predicting the OER activity and in forming an in-depth understanding of this complex reaction. We also note that catalysts which evolve $O_2$ through more intricate pathways could exhibit lower overpotentials than predicted by our analysis, although the extrapolation of this

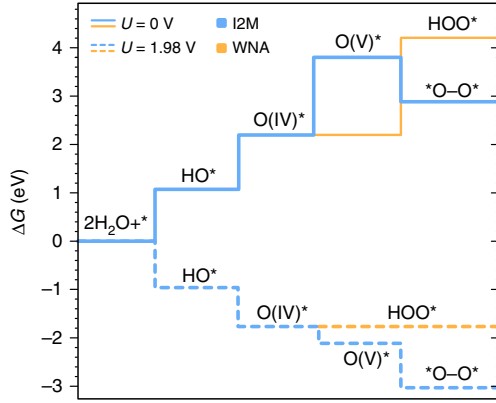

**Fig. 6** Gibbs energy diagram for **Ru-1** calculated at 0 V (solid lines) and 1.98 V (dashed lines) vs. RHE. Blue lines represent the I2M mechanism while the pale orange lines represent the conventional WNA mechanism

knowledge to the design of novel molecular catalysts is not straightforward. Instead, our work sets the foundations for the fast and efficient screening of molecular OER catalysts and posits that catalysts, which are close to the top of volcano as seen in Fig. 4b have the potential to evolve oxygen through both the WNA mechanism as well as through more complex routes. This approach, yet to be exploited in homogeneous catalysis, allows for the exploration of a substantial portion of the vast potential chemical space, which would be otherwise intractable, and hence, it is expected to lead to the discovery of novel catalysts with an improved performance in a reasonable time.

**Fundamental principles for the design of ideal molecular OER catalysts**. With all this knowledge, in the following we establish and discuss a series of catalyst design principles to accelerate the discovery of novel molecular systems based on earth-abundant elements exhibiting an ideal OER activity. Firstly, we propose that any high-throughput search should focus on transition metal complexes which can stabilize high oxidation states and thereby undergo the additional ET step to hurdle the OER overpotential wall. Although a complete characterization of the activity of these complexes would require the modeling of many more intermediates, the approach presented herein allows for the rapidly accelerated screening of molecular catalysts using thermodynamic OER descriptors. Special attention, however, should be paid when

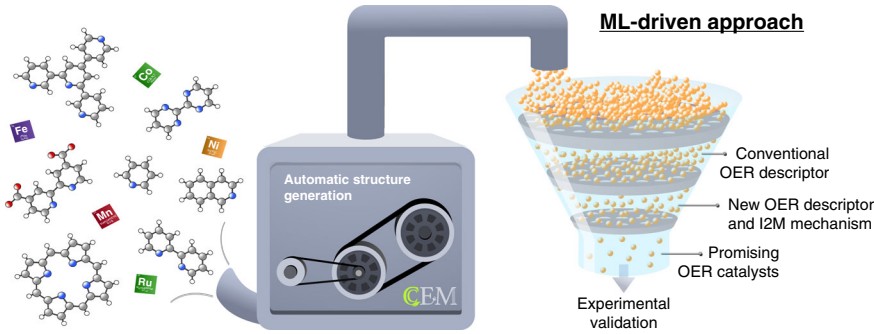

**Fig. 7** Proposed approach for the accelerated discovery of ideal OER catalysts

predicting the activity of catalysts with a low conventional OER descriptor—those lying on the left leg of the volcano in Fig. 4a— as their activity can be underestimated if they undergo an extra ET step. To prevent this, we propose to use the Gibbs energy change for the additional ET as a new OER descriptor. In addition, the I2M mechanism must also be taken into account for those complexes so as to avoid overestimating the overpotential by implicitly assuming the oxygen evolution proceeds via a WNA from the O* to HOO* intermediates.

Future design of highly active molecular OER catalysts may also benefit from the knowledge available in the heterogeneous catalysis community, where OER materials with reduced overpotentials have been successfully predicted by means of DFT calculations and scaling relations. An instructive example is the recent computer-aided design of a ternary Co, Fe, and W oxide material which exhibited a record low overpotential in alkaline electrolyte[18]. Therein, the optimal binding of the HO* intermediate of the ternary system was tailored by combining pure metal oxides and oxyhydroxides with weak and strong binding energies. By analogy, in molecular systems, we envisage that a similar effect could be achieved by altering the transition metal center to produce a favorable binding energy. The blending of features of two homogeneous catalysts is, however, a more challenging process. Particularly, we anticipate that replacing the metal center in molecular systems may lead to a more drastic effect on the electronic structure compared to doping of heterogeneous systems, where the effect is more distributed throughout the system. This may completely change the OER descriptor value and lead to a different PLS, illustrating the innate difficulty in tuning catalysts involving many ETs and intermediates.

In designing more efficient OER complexes, the choice of ligand is crucial. For example, H-bonding with the metal ligands can be leveraged to improve (or break) the OER scaling between the HO* and HOO* intermediates, while ligands with different electronic properties may play a role in stabilizing other high-valent intermediates or by acting as a proton shuttle to improve the overall reaction kinetics, as has been experimentally demonstrated recently[31]. Therefore, the electronic and steric effects of the ligand environment need to be systematically investigated to illuminate potential strategies for merging the desirable features of distinct ligands to identify highly active OER catalysts. Importantly, advances in this area would enable the future implementation of ideal molecular OER catalysts into commercial electrolyzers via their immobilization onto conductive electrode supports leading to hybrid materials with greatly enhanced selectivity, stability, and electron transport. In fact, this has been experimentally achieved for a wide variety of molecular systems[63–66] and theoretically proven to be feasible for model Ir catalysts with no apparent loss of activity[67], although substantial work remains to be done in order to devise more efficient and robust hybrid OER materials based on abundant elements.

Finally, one has to realize that homogeneous catalysts are essentially infinitely tunable owing to the vastness of chemical space[68]. Hence, highlighting the fruitful regions within this space is a most demanding and exciting challenge which we can begin to address with these design principles. With ample amounts of data, we also envision machine learning (ML) algorithms to become an integral part of such processes to greatly enhance the speed of catalyst discovery in an automatized approach, as illustrated in Fig. 7.

Overall, this work provides strong theoretical evidence that OER scaling relations can be generalized to any material regardless of their nature. Additionally, it demonstrates that the conventional OER descriptor can be applied to rapidly screen molecular catalysts and to predict their activity requiring only a few DFT calculations. This descriptor, however, underestimates the catalytic performance of certain metal complexes with the O* to HOO* transition as PLS, since it does not consider the additional one-electron oxidation that they undergo before O–O bond formation. To capture this behavior, we propose using a new OER descriptor based on the Gibbs energy of that extra oxidation, and demonstrate that it provides a good account for the catalytic performance of the most active complexes considered herein, according to their reported experimental TOFs. This proves the predictive power (both qualitative and semiquantitative) of the new OER descriptor, which is unprecedented. In addition, this new descriptor pinpoints the additional one-electron oxidation as the means by which certain catalysts can circumvent the "overpotential wall". In fact, we predict that molecular systems undergoing that extra step can exhibit zero theoretical overpotential provided the energy levels of the relevant OER intermediates are properly aligned. Finally, all the knowledge acquired in this work has been used to establish the fundamental principles for the rational design of ideal OER catalysts. We expect that these guidelines can be applied in a fully automated approach driven by ML algorithms and OER descriptors to accelerate the discovery of ideal OER materials based on earth-abundant elements. Altogether, this work represents a significant conceptual advance towards developing energy conversion and storage technologies based on water splitting by bringing together concepts from homogeneous and heterogeneous catalyses. Therefore, we foresee this work will inspire future research directions in the field and spur the development of commercial water electrolyzers.

## Methods

DFT calculations were performed at the TPSSh level using the Gaussian09 software[69]. The choice of this functional was based on the satisfactory results obtained in a preliminary benchmark study of the redox potentials of four molecular OER catalysts investigated in this work using three different functionals with varying levels of Hartree–Fock (HF) exchange, namely B3LYP (20% HF)[70], TPSSh (10% HF)[71,72], and BP86 (0% HF)[73,74]. The results and details of these calculations are presented in Supplementary Table 1. Overall, the TPSSh functional provided the lowest average

MAE (MAE = 303 mV) compared with the reported experimental values (see Supplementary Table 1), and therefore, it was chosen for subsequent theoretical investigations. We note that an error of ca. 300 mV is within the accuracy reported in previous theoretical works[55]. This choice of functional, in concert with the chosen basis sets is further supported by the satisfactory results obtained in the computation of heats of formation of many different transition metal complexes[75].

To describe the Ru, Mn, Fe, Cu, Co, and Ni metals, the Lanl2dz effective core potential was used along with $f$-polarization functions with exponents 1.235, 2.195, 2.462, 3.525, 2.78, and 3.13, respectively[76,77]. The more electronegative O, N, F, and Cl atoms were described using the 6−31+G(d) basis set, while the 6−31G(d,p) basis set was employed for C and H atoms. To account for non-covalent interactions, Grimme-D3 dispersion corrections[78] were added to the Gibbs energy through single point calculations using the optimized geometries in solution (see below). We note that the primary conclusions of this work remain the same regardless of the addition of dispersion; data for calculations without dispersion corrections are included in the Supplementary Figs. 3 and 4. For catalysts with two potential active sites, only one was considered for simplicity. This approach has been successfully applied to the modeling of the OER activity of polyoxometalates[79,80], providing results in good agreement with experiments.

Molecular structures were optimized in solution using the implicit SMD solvation model with $H_2O$ ($\varepsilon = 78.3553$) as a solvent[81]. Whenever high-spin and low-spin states were accessible, both were considered in geometry optimizations and the lowest energy structure was selected.

Gibbs energies were calculated at the temperature of 298.15 K and pressure of 1 atm, except for the isolated $H_2O$ molecule that was computed at the temperature and pressure at which both the liquid and gas phases are in equilibrium, i.e. 300 K and 0.035 atm. Relative Gibbs energies are referenced to $H_2O$ and $H_2$ in solution, to avoid introducing the error associated with the modeling of $O_2$ with DFT methods, and the global reaction Gibbs energy was fixed to the experimental value of 4.92 eV. All the optimized structures were verified to be real energy minima by means of vibrational frequency analysis.

## Data availability

DFT calculations, structures, and related information can be found at the following online dataset: https://iochem-bd.bsc.es/browse/review-collection/100/193468/17b443753bc31a6f429b8e67.

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

## Acknowledgements

Authors gratefully thank Trinity College Dublin for financial support. Particularly, M.J.C. and E.M.-T. acknowledge funding from the Provost's Ph.D. Project Awards and Ussher Postgraduate Scholarships, respectively, generously funded through alumni donations and Trinity's Commercial Revenue Unit. The DJEI/DES/SFI/HEA Irish Centre for High-End Computing (ICHEC) is also acknowledged for the generous provision of computational facilities and support. J.S.-L. acknowledges funding received from the European Union's Horizon 2020 research and innovation program under the Marie Skłodowska-Curie grant agreement No. 713567. J.S.-L. and W.S. also acknowledge funding received from the European Research Council (CoG 2014-647719) and the Science Foundation Ireland (SFI 13/IA/1896).

## Author contributions

M.G.-M. designed and supervised the work. M.J.C., G.C., and E.D. carried out the DFT calculations with the assistance of J.S.-L. and E.M.-T. The manuscript was written by M.J.C. and M.G.-M. with the assistance of W.S. All authors discussed the results and assisted in the preparation of the manuscript.

## Competing interests

The authors declare no competing interests.
