## [Peer Review File · Nature Communications]

Reviewers' comments:

Reviewer #4 (Remarks to the Author):

I thank the authors for taking some time to read my comments and the references therein. The rebuttal is balanced and several aspects are now clearer to me. Before I gladly recommend publication in Nature Communications, I must say that the work by Hellman et al remained uncited in spite of its relevance.

Additionally, I do not agree with the answer to the last comment of my previous report. The offset of the OH-OOH scaling relation is 3.2 eV and has an uncertainty of ± 0.2 eV (according to Man et al), so when it is divided by $2e^-$ and 1.23 V are subtracted to obtain the theoretical top of the volcano of 0.37 V, the uncertainty associated to the latter number is still ± 0.2 V, not 0.1 V as the authors said.

Because of that, one can have overpotentials at the top of up to 0.17 V, i.e. $0.37 \text{ V} - 0.2 \text{ V}$. I made that particular comment initially and emphasize it again because conventional catalysts following the conventional pathway can have overpotentials as low as 0.17 V without having to break or workaround scaling relations. Thence, I kindly suggest that the concept of an "overpotential wall" be put in context with its associated nuances.

Reviewer #4 (Remarks to the Author):

I thank the authors for taking some time to read my comments and the references therein. The rebuttal is balanced and several aspects are now clearer to me.

Before I gladly recommend publication in Nature Communications, I must say that the work by Hellman et al remained uncited in spite of its relevance.

We thank the reviewer for their thoughtful, constructive feedback and we are very pleased by their satisfaction with our manuscript, as well as our recent replies. We hope that through these short comments they will recommend the publication of our revised manuscript.

Indeed, we agree that the work from Hellman *et al.* (*J. Am. Chem. Soc.* **2014** 136, 1320), helpfully provided by the reviewer, is relevant as it considers scaling relations for a set of functionalized porphyrins. This paper was initially omitted because we felt that in the introduction we had adequately described the breadth of investigations into molecular scaling relations for distinct classes of molecules. However, we now respect, as the reviewer mentions, that this paper is pertinent to our work, especially due to the further investigation into the effect of the second coordination sphere on hydrogen bonding.

Alterations to manuscript: Page 11, line 11: *"H-bonding can also be leveraged to boost the kinetics of the O* to HOO* step in the WNA mechanism, as recently shown by experiments,⁴⁹ while theoretical investigations have highlighted the possibility that the second coordination sphere can modulate the binding energy of the crucial O* intermediate.^{50"}*

Additionally, I do not agree with the answer to the last comment of my previous report. The offset of the OH-OOH scaling relation is 3.2 eV and has an uncertainty of ± 0.2 eV (according to Man et al), so when it is divided by $2e^-$ and 1.23 V are subtracted to obtain the theoretical top of the volcano of 0.37 V, the uncertainty associated to the latter number is still ± 0.2 V, not 0.1 V as the authors said. Because of that, one can have overpotentials at the top of up to 0.17 V, i.e. $0.37\text{ V} - 0.2\text{ V}$. I made that particular comment initially and emphasize it again because conventional catalysts following the conventional pathway can have overpotentials as low as 0.17 V without having to break or workaroud scaling relations. Thence, I kindly suggest that the concept of an "overpotential wall" be put in context with its associated nuances to the last comment of my previous report.

We thank the reviewer for continuing this interesting consideration. While this stimulating discussion does not affect the ultimate conclusions or insights drawn from our work, we wish to outline how we have calculated the uncertainty in question for clarity.

According to classical uncertainty propagation, relative uncertainties do not change when variables are divided (or multiplied) by a constant *with no uncertainty*, therefore, absolute uncertainties **must** change when divided (or multiplied) by a constant. In particular, those uncertainties will be divided (or multiplied) precisely by the constant dividing (or multiplying) the variable. In our case $(3.2 \pm 0.2)/2 = 1.6 \pm 0.1$. Hence, we contend that the absolute uncertainty associated with the "peak" of the volcano presented in Figure 3, does not remain ± 0.2 V but changes to ± 0.1 V. However, upon revisiting the paper by Man *et al.* (*ChemCatChem*, **2011**, 3, 1159), we note that 95 % of the reported catalysts were within ± 0.4 eV, and therefore the reviewer's overpotential wall of 0.17 V would hold in those special catalysts that deviate from the scaling by -0.4 eV. Hence, upon further reflection, we respect this point and have contextualized the initial discussion of the overpotential wall, as detailed below.

Alterations to manuscript: Page 6, line 12: *"While the uncertainty associated with the calculation of $\Delta G_{HOO^*} - \Delta G_{HO^*}$ can reduce that overpotential wall, as has been experimentally observed for a few exceptional OER electrocatalysts,¹⁵⁻²⁰ hurdling or reducing this wall remains a major challenge."*

REVIEWERS' COMMENTS:

Reviewer #4 (Remarks to the Author):

I still do not agree with the answer provided by the authors. I do not believe that at the crossing point of two lines (i.e. the volcano apex) each of which have a systematic error of ± 0.2 eV, the error is magically divided by one half. I kindly invite the authors to make the exercise of drawing the two lines with the error bands around them and verifying whether the error is one half at the crossing point compared to the rest of the lines. Without further ado, I leave it up to the editor to finally accept the paper, as I have already expressed my positive opinion about this article but my disagreement with the fact that the "overpotential wall" is not presented throughout the text with the wide error bars it has.